# Feasibility and Acceptability of Accelerometer Measurement of Physical Activity in Pregnant Adolescents

**DOI:** 10.3390/ijerph18052216

**Published:** 2021-02-24

**Authors:** Emma L. M. Clark, Lauren D. Gulley, Allison M. Hilkin, Bonny Rockette-Wagner, Heather J. Leach, Rachel G. Lucas-Thompson, Marian Tanofsky-Kraff, Kristen J. Nadeau, Stephen M. Scott, Jeanelle L. Sheeder, Lauren B. Shomaker

**Affiliations:** 1Department of Human Development and Family Studies, Colorado State University, Fort Collins, CO 80523, USA; emma.clark@colostate.edu (E.L.M.C.); lauren.gulley@childrenscolorado.org (L.D.G.); lucas-thompson.rachel.graham@colostate.edu (R.G.L.-T.); 2Department of Pediatrics, University of Colorado School of Medicine and Children’s Hospital Colorado, Aurora, CO 80045, USA; allison.hilkin@childrenscolorado.org (A.M.H.); kristen.nadeau@childrenscolorado.org (K.J.N.); stephen.scott@cuanschutz.edu (S.M.S.); jeanelle.sheeder@cuanschutz.edu (J.L.S.); 3Department of Epidemiology, Graduate School of Public Health, University of Pittsburgh, Pittsburgh, PA 15261, USA; bjr26@pitt.edu; 4Department of Health and Exercise Science, Colorado State University, Fort Collins, CO 80523, USA; heather.leach@colostate.edu; 5Community and Behavioral Health, Colorado School of Public Health, Aurora, CO 80045, USA; 6Department of Medical and Clinical Psychology, Uniformed Services University of the Health Sciences, Bethesda, MD 20814, USA; marian.tanofsky-kraff@usuhs.edu; 7Department of Obstetrics and Gynecology, University of Colorado School of Medicine, Aurora, CO 80045, USA

**Keywords:** at-risk, minorities, underserved populations, pregnancy measures, teen pregnancy

## Abstract

During pregnancy, physical activity relates to better maternal and child mental and physical health. Accelerometry is thought to be effective for assessing free-living physical activity, but the feasibility/acceptability of accelerometer use in pregnant adolescents has not been reported. In this short communication, we conducted secondary analysis of a small pilot study to describe the feasibility/acceptability of accelerometry in pregnant adolescents and the preliminary results of physical activity characteristics. Participants were recruited from a multidisciplinary adolescent perinatal clinic. Physical activity was assessed with wrist-worn accelerometers. Feasibility was described as median days of valid wear (≥10 h of wear/day) for the total sample and the number/percentage of participants with ≥4 days of valid wear. Sensitivity analyses of wear time were performed. Acceptability ratings were collected by structured interview. Thirty-six pregnant (14.6 ± 2.1 gestational weeks) adolescents (17.9 ± 1.0 years) participated. Median days of valid wear were 4 days. Seventeen participants (51.5%) had ≥4 days of valid wear. There were no differences in characteristics of adolescents with vs. without ≥4 days of valid wear. Twenty participants (60.6%) had ≥3 days of valid wear, 24 (72.7%) ≥2 valid days, and 27 (81.8%) ≥1 valid wear day. Acceptability ratings were neutral. Assessing physical activity with accelerometry in pregnant adolescents was neither feasible nor acceptable with the current conditions. Future research should investigate additional incentives and the potential utility of a lower wear-time criterion in pregnant adolescents.

## 1. Introduction

Physical activity (PA) during pregnancy is as an important contributor to maternal and child health outcomes [1]. Pregnant adults who engage in routine PA are more likely than those who are sedentary to have better mood, healthier cardiovascular function, decreased risk of developing gestational diabetes, and less postpartum weight retention [2], and to give birth to infants with healthy weight and lower risk of childhood obesity [3]. Although PA during adult pregnancy has been extensively studied [4,5,6], investigations into PA during adolescent pregnancy are highly limited, despite this group’s increased risk for inadequate PA and excess gestational weight gain [7,8]. Pregnancy in adolescence can exacerbate increases in fat deposition that normally characterize puberty and pregnancy [9], which may increase adolescents’ risk for excess gestational weight gain and the continued intergenerational transmission of obesity and cardiometabolic disease [10]. The American College of Obstetrics and Gynecologists recommends at least 30 min per day of moderate-to-vigorous PA (MVPA) during pregnancy [1]. Higher total daily PA has been associated with less excess gestational weight gain, lower risk of diabetes, and less postpartum weight retention in pregnant women [11]. Despite the importance of PA in pregnancy, there are no studies objectively quantifying PA using accelerometry in pregnant adolescents.

Instruments that objectively quantify PA, such as accelerometers, reduce self-report biases, leading to more accurately measured PA in pregnant adults and non-pregnant adolescents [12,13]. However, the social–emotional and behavioral difficulties that often complicate adolescent pregnancy likely have contributed to this group being understudied with respect to PA [14]. For example, serious adverse life events, including abuse, domestic violence, poverty, and homelessness, unfortunately are common in pregnant adolescents, and rates of depression are high (16−44%) [14,15]. Although high-quality data collection using accelerometry with economically disadvantaged adult women is feasible [16], it is unknown whether the additional social–emotional and behavioral challenges of pregnant adolescents could undermine PA instrument implementation in adolescent pregnancy. Therefore, research that seeks to investigate and establish accurate PA assessment in this high-risk, high-need group is a necessary foundation for future research. 

In the current short communication, we conducted secondary data analysis of a pilot study at baseline to characterize accelerometry feasibility and acceptability in first-trimester, pregnant adolescents. We predicted that pregnant adolescent PA measured with accelerometry would be feasible such that valid wear (≥10 h wear/day) would be equal to or exceed a median of 4 days, and that at least 60% of participants would demonstrate ≥4 days of valid wear [17,18]. We also predicted that accelerometry would be acceptable to this population, such that participants would endorse above-average scores on liking and ease (≥4 out of 5), suggesting that wearing the device was acceptable.

## 2. Materials and Methods

### 2.1. Design and Participants

This short communication utilizes secondary data from the baseline phase of a small pilot trial testing a behavioral intervention for improving mood and reducing excess gestational weight gain [19]. Inclusion criteria were (i) 13–19-years-old, (ii) pregnant, 12–18 gestational weeks, and (iii) receipt of prenatal care at an adolescent perinatal clinic. Exclusion criteria were (i) current full-syndrome psychiatric disorder that could interfere with study participation, such as conduct disorder, schizophrenia, substance abuse, or active suicidal ideation, (ii) regular medication use affecting mood/weight, such as anti-depressants, (iii) high-risk pregnancy medical conditions, such as preeclampsia, gestational diabetes, hypertension, or multiple gestation, (iv) major renal, hepatic, or endocrinological disorders, such as hyperthyroidism, or a pulmonary disorder other than mild asthma, and (v) pre-pregnancy body mass index (BMI) < 5th percentile for age/sex.

### 2.2. Process

Adolescents participated in a screening/baseline visit. Research staff obtained written, active informed consent. According to Colorado State Law, a pregnant minor is emancipated to approve prenatal, delivery, and postdelivery medical care for herself in relation to the intended live birth of a child (Colorado Revised Statute 13-22-103.5) and may consent for herself. Participants were fitted with an accelerometer on the non-dominant wrist and directed to wear the device at all times for 7 consecutive days/nights, other than immersive water-based activities [17,18]. The device was collected at a return visit or prenatal appointment. Participants reported acceptability by interview. Due to the nature of the pilot study, adolescents were compensated for screening/baseline visit participation regardless of degree of accelerometer-wear compliance.

### 2.3. Measurements

PA was measured with ActiGraph’s GT3X accelerometer worn on the non-dominant wrist, which has been validated in non-pregnant adolescents, non-pregnant adults, and pregnant adults [20,21,22]. Although PA may vary weekly, this measurement has demonstrated good week-to-week reliability for habitual PA when measured over a one to two-week period [22]. Wear-time compliance was obtained through the ActiLife software [20].

Feasibility was operationalized as days of valid wear (e.g., ≥10 h wear/day) and number/percentage of participants with valid wear. Feasibility was defined as median ≥4 days of valid wear (including a minimum of one weekend day with valid wear time) and 60% of participants with ≥4 days of valid wear; these thresholds were based upon Van Coevering and colleagues’ [23] and Hesketh and colleagues’ [21] studies using accelerometers to measure PA with non-pregnant adolescents and pregnant adults, respectively.

Acceptability was measured by ratings on an adapted, structured acceptability interview. Likeability/acceptability items were rated on a Likert scale from 1 = Totally Disagree to 5 = Totally Agree with questions including, “I liked wearing the activity watch” and “Wearing the watch got easier with time and practice”. Acceptability was operationalized as an average score of ≥4 out of 5, indicating that wearing the device was likeable/acceptable. Adolescents selected from a checklist of reasons for wearing the accelerometer (e.g., “helping the study”, “keeping track of my physical activity”) and reasons for not wearing the accelerometer (e.g., “forgetting to wear it”, “it was uncomfortable”). Additionally, adolescents selected from a checklist of factors that would improve wear (e.g., “more reminders about the watch”, “getting extra money for wearing the watch”).

Participants reported their height/weight prior to pregnancy to estimate pre-pregnancy BMI [24]. First-trimester BMI, BMI *z*-score, and BMI percentile were calculated based on participants’ measured weight (kg) divided by their measured height (m) squared [25]. Height was measured three times to the nearest millimeter by trained research staff using a stadiometer. Weight was measured to the nearest 0.1 kg by a calibrated electronic scale.

Adolescent age, race/ethnicity, gestational weeks, and other medical/psychiatric information were obtained from the medical chart. These characteristics were descriptive (e.g., race/ethnicity) and/or necessary in order to determine eligibility.

### 2.4. Data Analysis

Raw data from the accelerometers were downloaded using ActiLife.6 software and used to convert raw data, originally collected in 1 s epochs, into accelerometer count-based data that were then summed into 60 s epochs (time sampling interval) with at least 20 min of continuous counts of 0 intensity excluded from the analysis [20]. As the optimal cut-point for pregnant adolescents has not yet been established, Freedson [26] cut-points from ActiLife.6 were used to identify time spent in different intensities of PA with 100 counts per minute for the sedentary cut-point. These cut-points have been used in samples of pregnant adult women using waist-worn accelerometry [20,21,22]. There is no standard processing criterion to determine activity intensity for wrist-worn accelerometers in pregnant adults; therefore, we used Actigraph’s protocol for assessing physical activity with wrist-worn accelerometers. We used the ActiLife.6 software’s Freedson [26] cut-points in conjunction with the selection of the option “worn on wrist” and hand non-dominance indicated in order for the software to appropriately scale the selected cut-points accounting for the wrist. Wear-time was obtained through the ActiLife software [20]. Sensitivity analyses were performed using 1–3 days to compare against the ≥4 valid days criterion, in order to investigate whether these cut-points yielded different feasibility. Independent samples t-tests and chi-squared compared adolescents who did and did not have ≥4 valid days on demographics and BMI indices. Descriptive and frequency statistics were utilized to calculate Mean ± SD likeability/ease of watch wear, number/percentage of endorsed reasons for wearing or not wearing the accelerometers, and number/percentage of factors that would improve wear. Among those with valid data, descriptive statistics were generated on 4 PA dimensions: (i) average minutes MVPA/day, (ii) average minutes sedentary bouts/day, (iii) average number steps/minute, and (iv) maximum number steps/minute [26].

## 3. Results

Baseline characteristics of 36 pregnant adolescent participants are provided in Table 1. Median valid days were 4 (interquartile range 1–7; 95% CI [2.60, 5.28]). Thirty-three participants (91.7%) returned the device. Of those who returned the device, 17 (51.5%) had ≥4 days of valid wear (95% CI [34%, 70%]). Twenty-seven participants (81.8%) had ≥1 valid wear day (95% CI [68%, 96%]), 24 participants (72.7%) had ≥2 valid wear days (95% CI [57%, 89%]), and 20 participants (60.6%) had ≥3 days of valid wear (95% CI [43%, 78%]). There were no differences in race/ethnicity, age, or gestational weeks for those with and without valid wear, analyzed at ≥4 days. Compared to adolescents with ≥4 days of valid wear, those with <4 days showed a non-significant trend toward higher self-reported pre-pregnancy BMI (26.84 ± 6.14 vs. 23.68 ± 5.07 kg/m^2^, *p* > 0.05), with no other difference.

Seventeen participants (47.2%) completed the acceptability interview. Hispan-ic/Non-Hispanic Black adolescents were more likely to respond to the interview than participants of other races/ethnicities (*p* < 0.05). There was no difference in accelerom-eter wear days for those who did and did not complete the interview. Of the partici-pants who did not return the device, three completed the acceptability interview. Par-ticipants endorsed an average of 2.70 ± 1.00 regarding to what extent they liked wear-ing the accelerometer (Scale: 1−5) and 3.20 ± 1.50 for wearing the accelerometer be-coming easier with practice (Scale: 1−5). Two participants (11.8%) liked wearing the accelerometer (i.e., Likert-score ≥4 out of 5), and nine (52.9%) reported that wear be-came easier with practice (i.e., ≥4 out of 5). Desire to comply with the study protocol was the most common reason (52.9%) for wearing the accelerometer. The main reason for non-wear was forgetting (64.7%). Participants endorsed more frequent reminders (76.5%), enhanced comfort (76.5%), and more monetary incentives (64.7%) as factors that would improve wear.

Among pregnant adolescents with ≥4 days of valid wear (*n* = 17), descriptive in-formation for PA patterns is presented in Table 2. Thirteen adolescents (76.5%) met or exceeded the recommended guidelines of 30 min of MVPA/day.

## 4. Discussion

This short communication describes the first assessment of feasibility and acceptability of accelerometry to measure PA in pregnant adolescents, a group at high risk for perinatal mental health difficulties and excess gestational weight gain [14]. Accelerometry was neither feasible nor acceptable, highlighting important differences compared to non-pregnant adolescents and pregnant adults [21,23]. In contrast to accelerometry feasibility compliance in non-pregnant adolescent samples (80%) [23] and pregnant adult samples (74%) [21] without monetary wear time incentives, only 47.2% (*n* = 17 of 36) of the pregnant adolescent participants in this pilot study had valid data. Given that our criterion for accelerometer feasibility was operationalized as at least 60% of participants would demonstrate ≥4 days of valid wear (≥10 h wear/day), accelerometry did not demonstrate feasibility in this pilot study.

Investigation into sensitivity analyses suggested that while less than half of participants had ≥4 days of valid wear (47.2%), a majority (60.6%) had ≥3 days of valid wear. Although this analysis was performed for descriptive purposes only and extant research suggests a 7-day measurement period in other samples [21,27], future studies may benefit from the exploration of various criterion thresholds in this population with compliance challenges.

Further contrasting previous studies with non-pregnant adolescent and pregnant adult samples that report relatively high acceptability of accelerometry [23,28], this sample of pregnant adolescents reported “neutral” as opposed to high acceptability regarding accelerometer wear. These neutral ratings did not meet our “above-average” acceptability criterion (i.e., ≥4 out of 5), suggesting that wearing the device was not acceptable in the current sample. Pregnant adolescents represent a high-risk group typically facing significant social–emotional and behavioral health challenges, in addition to socioeconomic disadvantages [14,15], which may have undermined adherence. However, adolescents reported that opportunities to re-wear the device, more frequent reminders, and monetary incentives would increase acceptability, suggesting that obtaining more usable accelerometer data in future studies may be possible if additional measures are taken to increase compliance. Testing such approaches should be undertaken in future studies with pregnant adolescents.

Of pregnant adolescents with valid wear, ~75% met or exceeded MVPA recommendations, exceeding estimates in pregnant adults or self-reported PA in pregnant adolescents [29]. Although this estimate may be influenced by participant non-response bias, it is also possible that physically intensive work or labor and/or reliance on public transportation could have contributed to higher MVPA. Future studies should distinguish leisure- and non-leisure-time PA to better understand these contextual factors.

Strengths of this report include objective assessment of PA via accelerometry in a racially and ethnically diverse sample of pregnant adolescents. Accelerometry offers an important, objective evaluation of the length and intensity of PA and sedentary time. Pregnant adolescents are at heightened risk for insufficient PA as well as excess gestational weight gain and its downstream consequences for obesity and cardiometabolic disease, making the study of PA important in this high-risk, underserved group. Although not exhaustive, key shortcomings of this study include that, due to the limited financial resources of this pilot study, participants were drawn from a small convenience sample, limiting generalizability. Moreover, participants could not be incentivized for accelerometer-wear specifically. The sample size was not powered to test PA associations with health characteristics nor statistically significant differences between individuals with valid wear compared to those without valid wear, which might have masked differences (e.g., in pre-pregnancy BMI). There is lack of consensus regarding optimal placement of accelerometers in pregnancy. Waist wear may be more accurate than wrist wear but can be uncomfortable and monitors may slip in pregnancy. Pregnant women are often counseled to exercise moderately between weeks 20 and 37 to maintain good health and to prepare for childbirth [30,31]; pregnant adolescents in the current study were assessed earlier in pregnancy (week 15), which limits the generalizability of the results. Finally, given the limitations of this small pilot study’s size and scope, future research on accelerometer use in larger samples of pregnant adolescents is warranted.

## 5. Conclusions

Although the objective assessment of PA via accelerometry in this racially and ethnically diverse sample of pregnant adolescents was neither feasible nor acceptable, this study makes an incremental contribution to the literature in developing and improving the use of this methodology in this high-risk population where traditional accelerometry methodology may not be feasible. Further investigation into these negative results is essential for developing alternative and innovative strategies and guidelines around accelerometry that effectively characterize PA in this underserved population. The combination of psychosocial challenges and excess gestational weight gain risk may render pregnant adolescents particularly vulnerable to adverse health outcomes, yet PA may be a potentially modifiable factor that offers the potential to increase perinatal emotional wellbeing and weight/metabolic health outcomes.

## Figures and Tables

**Table 1 ijerph-18-02216-t001:** Descriptive information for study participants.

Characteristic	Total Sample *n* = 36	Adolescents with Valid Wear ^a^ *n* = 17	Adolescents without Valid Wear *n* = 19	
	Mean ± SD or % (*n*)	Mean ± SD or % (*n*)	Mean ± SD or % (*n*)	*p*
Age, years	17.86 ± 1.02	17.82 ± 1.07	17.89 ± 1.00	0.84
Race/ethnicity				0.97
Hispanic	36.1% (13)	35.3% (6)	36.8% (7)	
Non-Hispanic Black	38.9% (14)	35.3% (6)	42.1% (8)	
Non-Hispanic White	8.3% (3)	11.8% (2)	5.3% (1)	
Asian	5.6% (2)	5.9% (1)	5.3% (1)	
Multiple	11.1% (4)	11.8% (2)	10.5% (2)	
Gestational age, weeks	14.60 ± 2.10	14.76 ± 2.55	14.45 ± 2.34	0.71
Pre-pregnancy BMI ^b^, kg/m^2^	25.35 ± 5.81	23.68 ± 5.07	26.84 ± 6.14	0.10
Pre-pregnancy weight status				0.25
Lean, BMI 5−84th percentile	61.1% (22)	70.6% (12)	52.6% (10)	
Overweight, BMI 85–94th percentile	22.2% (8)	23.5% (4)	21.1% (4)	
Obese, BMI ≥95th percentile	16.7% (6)	5.9% (1)	26.3% (5)	
First-trimester BMI, kg/m^2^	25.93 ± 6.39	24.78 ± 6.17	26.95 ± 6.58	0.32

^a^ Valid wear refers to ≥4 days of ≥10 h/day of wear. ^b^ BMI = body mass index.

**Table 2 ijerph-18-02216-t002:** Physical activity indices of pregnant adolescents with valid wear (*n* = 17).

Variable	M ± SD	Range
Average MVPA ^1^ in minutes per day	108.16 ± 59.71	2.90–188.20
Average sedentary bouts in minutes per day	414.72 ± 210.08	148.30–740.80
Average step counts per minute	7.15 ± 2.64	0.60–11.60
Maximum step counts per minute	100.00 ± 25.57	21.00–127.00

^1^ MVPA = moderate-to-vigorous physical activity.

## Data Availability

Data are available upon reasonable request.

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
