# Peer review of "Feasibility and Acceptability of Accelerometer Measurement of Physical Activity in Pregnant Adolescents"

_ijerph, 2021, doi:10.3390/ijerph18052216_

Round 1
Reviewer 1 Report
Feasibility and acceptability of accelerometer measurement of physical activity in pregnant adolescents
The primary purpose of the study is to assess feasibility and acceptability of wearing an accelerometer to measure physical activity among 36 pregnant adolescents. Secondary analysis estimates the association between physical activity and depression within this same cohort.
Primary conclusions state that only 51.5% of participants had valid wear, and acceptability of wearing the device was ‘neutral’. Average MVPA/day was inversely associated with depression scores. Authors suggest investigating the validity of a lower wear-time criterion in pregnant adolescents.
Conclusions seem generally appropriate, but I suggest being more forthright and just state that this study found that the accelerometer was not feasible or acceptable in this cohort of participants.
A major limitation of this study is sample size. There were no a priori sample size calculations. Analysis using a non-inferiority type of analysis may have provided a better approach to assessing feasibility. For example, the null hypothesis would assume the proportion of participants with valid wear is <=0.60+moni and the alternative is >=0.60+moni. Where moni is the margin of non-inferiority.
However, the analytic approach presented is ok.
Please add confidence intervals for primary outcomes (i.e., % with valid wear and median valid days).
There were no statistically significant differences between those with valid wear and those without. However, BMI was about 3 units greater in the non-valid wear group. Limitation of sample size.
Another limitation is that only 33/36 (91.7%) returned the device and 17/36 (47%) completed the acceptability interview. Did any of the participants who did NOT return the device participate in the acceptability interview?
The correlation between MVPA/day and depression is interesting. I suggest including a figure with depression scores on yaxis and MVPA/day on xaxis. Scatterplot (bubble plot?) with regression line.
The conclusions in section 5 should be deleted and replace with similar conclusions that are in the abstract. The conclusions are that this device was not feasible and acceptable. Understanding why would be important future work.
Author Response
Thank you very much for the excellent suggestions. We have responded to each comment in the attached document.

Reviewer 2 Report
Regarding the originality of the study, this type of analysis has been carried out for a long time and there is scientific documentation that proves it. The results of this work confirm what other authors have already published.
Accelerometry, is shown as one of the most reliable techniques, in the recording and storage of the amount and level of physical activity, performed by each person and in a given period of time. Studies in pregnant women show reliable data. The literature analyzed establishes accelerometry as an effective method to assess activity.
The authors make a brief communication in which they use data from a previous pilot test. This could be inconvenient.
The short communication on Accelerometry is presented by 11 authors. I consider the number of authors excessive and although the contribution of each one of them is specified, the work developed individually is not clear. I consider the article and number of authors disproportionate.
All the keywords proposed by the authors are already in the title. These must be replaced.
The scientific literature (Ojiambo et al. 2011; Hart et al. 2011; Sirard et al. 2011) is clear in stating that the mean measurement period with accelerometry must be a minimum of 7 days and a maximum of 8.5 days to obtain reliable values. The subject must wear the accelerometer at all times, except when sleeping or performing water activities. On those days, weekdays and weekends should also be included, since great differences have been found between one and the other.
Ojiambo R, Cuthill R, Budd H, Konstabel K, Casajús JA, González-Agüero A, Anjila E, Reilly JJ, Easton C, Pitsiladis YP; IDEFICS Consortium. Impact of methodological decisions on accelerometer outcome variables in young children. Int J Obes (Lond) 2011; 35 (Suppl 1): S98-103.
Hart TL, Swartz AM, Cashin SE, Strath SJ. How many days of monitoring predict physical activity and sedentary behavior in older adults? Int J Behav Nutr Phys Act 2011 16; 8:62.
Sirard JR, Forsyth A, Oakes JM, Schmitz KH. Accelerometer test-retest reliability by data processing algorithms: results from the Twin Cities Walking Study. J Phys Act Health 2011; 8 (5): 668-74.
The main drawback of the study is that of 36 pregnant adolescent participants, 33 (91.7%) returned the device. Of those who returned the device, only 17 (51.5%) had ≥4 days of valid use. Most of the participants, 81.8%, only had ≥1 days of valid use.
The exact days that pregnant women kept the heart rate monitor are not specified, as the authors themselves recommended that pregnant women keep it more than 7 nights and days in a row.
Another drawback is that exercise should be done moderately between weeks 20 and 37, to maintain good health in the pregnant woman and to prepare for childbirth.
Fernandez-Martinez O, Bueno-Cabanillas A, Martinez-Martinez M, Jimenez-Moleon JJ, de la Higuera MJL. Validity and reliability of a physical activity questionnaire for pregnant women. Archives of Medicine 2008; Four. Five).
Evenson KR, Calhoun KC, Herring AH, Pritchard D, Wen F, Steiner AZ. Association of physical activity in the past year and immediately after in vitro fertilization on pregnancy. Fertil Steril 2014 Feb 10. pii: S0015-0282 (13) 03477-8.
The adolescents studied are in week 15.
The authors have not indicated whether episodes of continuous 20 minutes with counts of intensity 0 were excluded from the analysis, considering these periods, the time without burnout.
Adolescents completed the 10-item Edinburgh Postnatal Depression Scale, validated in pregnant adolescents. The authors do not refer in Material and Method to the article in the
British Journal of Psychiatry (1987), 150, 782-786. Detection of Postnatal Depression Development of the 10-item Edinburgh Postnatal Depression Scale by J. L. COX, J. M. HOLDEN and R. SAGOVSKY,
nor is it listed in the literature.
A postnatal depression scale is used in the prenatal period.
The high adherence to accelerometer monitoring of economically disadvantaged women demonstrates that high-quality data collection in these populations is possible.
Sharpe PA, Wilcox S, Rooney LJ, Strong D, opkins-Campbell R, Butel J, Ainsworth B, Parra-Medina D. Adherence to accelerometer protocols among women from economically disadvantaged neighborhoods. J Phys Act Health 2011; 8 (5): 699-706.
The advantage of the accelerometer is that it is possible to evaluate how long a person is doing physical activity at different intensities and the sedentary lifestyle itself, which, when conducting research in this area, is very useful.
The discussion is very poor, it must be completed.
I consider that the work has many gaps that must be resolved.
Author Response
We greatly appreciate the reviewer's feedback and the opportunity to revise our manuscript in response to the suggestions. We have addressed each of the reviewer's comments and noted our response to each piece of feedback in the attachment document.

Reviewer 3 Report
Accelerometry is thought to be the gold-standard for assessing free-living PA. As you said, ‘gold-standard’, this sentence should be revised.
Replace “depression symptoms” with “depressive symptoms”.
Lack a comma in the bracket (r=-.55 p=.02).
Do not use p = XX, should be p < XX.
Line 35, should cite some references.
Line 46-47 needs to cite some references.
The structure of methods is not well-designed, please keep in line with some standard, like Study design, measurements, statistical analysis.
However, in the discussion, it seemed you failed to analyse the feasibility and acceptability based on your results. And another issue is that I did not see evidence to support feasibility. Why did you say “Feasibility was described as median days of valid wear”.
Based on the comments, I believe there is some major problems in the methodologies. A major revision is needed.
Author Response
We appreciate the feedback from the reviewer and have addressed each comment in the attachment document. We believe that the suggestions have substantially strengthened this short communication.

Reviewer 4 Report
With all the humility these recommendations are collected with the intention that they can be of help to improve this work.
1) Abstract
• Avoid acronyms that are not necessary in the abstract, for example Physical activity (PA). I think these acronyms in the body of the text would be more recommended.
2) Introduction
The introduction is very short, it is suggested to make a more in-depth review, since there are only six citations, which indicates that with only six studies the background and scientific evidence of the study topic has not been sufficiently collected.
It is recommended:
• Expand the benefits of practicing PA.
• Explain in greater depth what type of AF practice is the most recommended.
• Specify the benefits for the future baby that the mother is physically active.
• Develop more precisely the objectives and description of the working hypotheses.
3) Materials and Methods
• Explain the process of selecting the sample of participants.
• Distribute in the following sections: Design and Participants; Instruments; Process; Data analysis.
• Explain in more detail the inclusion and exclusion criteria to participate in this study.
The sample is very small, it is suggested to expand the sample. This is considered a very important handicap and one that must be resolved.
4) Results
• Although the results are clearly commented, I think other analyzes could be considered, such as: linear regression analysis.
5) Discussion
• This section must be improved. Only 4 investigations have been cited. A more extensive bibliographic review is suggested. In the current circumstances the discussion, like the introduction, has to be expanded and updated.
6) Conclusions
• In addition to what is contemplated, the strengths and limitations of the study should be reflected. Future prospects must be incorporated.
7) Bibliographic references
• It must be expanded and updated. It surprises me that a work of these characteristics and sent to a magazine of such prestige and with such a good impact factor, only has 12 citations.
• I have only found 3 quotes from the last 5 years.
Congratulations on your research.
Or, In the attached document you also can read all the suggestions.

Author Response
We appreciate the reviewer's thoughtful and constructive feedback. We have addressed each point in the attached response.

Round 2
Reviewer 2 Report
Highlight the effort made by the authors to improve communication.
We are facing a pilot study in which the authors only analyze whether or not they comply with the use of accelerometers to measure physical activity in pregnant adolescents. The results of this study are poor.
I still think that this short communication on Accelerometry is presented by eleven authors. I consider the number of authors excessive for the results offered, which are not conclusive.
I do not see clear the objective of the work in relation to the results obtained in the BMI. BMI assessment was not an objective of the study. The average measurement period with accelerometry should be a minimum of 7 days and a maximum of 8.5 days to obtain reliable values. The authors only specify that a total of 17 pregnant women (less than 50% of the total sample) had ≥4 days of valid use with the accelerometer. The authors acknowledge that only 47.2% of the sample had valid data.
The authors provide BMI data before pregnancy. It is not clear how these data were collected. They provide BMI data at the end of the first trimester of pregnancy without significant differences, but this is not the aim of the study as we have already commented.
Most of the participants, 81.8%, only had ≥1 days of valid use.
The adolescents studied are in week 15. To prepare for childbirth, physical activity must be performed between weeks 20 and 37. There are no data in these weeks.
The discussion is still poor.
I consider that the work has many gaps that must be resolved. It is only concluded that further investigation is essential, since the use of accelerometry was neither feasible nor acceptable in the population studied.
Bibliographic Citation No. 25 Incomplete
Van Hees, V. T .; Renstrom, F .; Wright, A .; Gradmark, A .; Catt, M .; Chen, K. Y .; Lof, M .; Bluck, L .; Pomeroy, J .; Wareham, N. J .; Ekelund, U .; Brage, S .; Franks, P. W., Estimation of daily energy expenditure in pregnant and non-pregnant women using a wrist-worn tri-axial accelerometer. Plos One 2011, 6, (7).
PLoS ONE 6 (7): e22922.
Put full number on the end page of the appointment. Example:
Hayes, L .; Bell, R .; Robson, S .; L, P .; UPBEAT Consortium, Association between physical activity in obese pregnant 261 women and pregnancy outcomes: the UPBEAT pilot study. Ann Nutr Metab 2014, 64, (3-4), 239-46.
Ann Nutr Metab 2014, 64, (3-4), 239-246.
All journals cited in the bibliography must be abbreviated. Example:
Shomaker, L. B .; Gulley, L. D .; Clark, E. L. M .; Hilkin, A. M .; Pivarunas, B .; Tanofsky-Kraff, M .; Nadeau, K. J .; Barbour, L. A .; Scott, S. M .; Sheeder, J. L., Protocol for a pilot randomized controlled feasibility study of brief interpersonal psychotherapy for addressing social-emotional needs and preventing excess gestational weight gain in adolescents. Pilot and Feasibility Studies 2020, 6, 39.
Pilot Feasibility Stud 2020, 6, 39.
Author Response
Thank you very much for the second round of feedback. We have responded to the best of our ability in the attached document.

Reviewer 3 Report
The authors have addressed my comments effectively.
Author Response
We are very pleased that the reviewer found our revised manuscript to be responsive to their suggestions. Thank you!
Reviewer 4 Report
I believe that the sections, which I indicated in the previous review, have been improved in a satisfactory way.
Author Response
Thank you for the positive feedback. We are glad the first round of revisions was responsive to the reviewer's feedback and resulted in a considerably strengthened manuscript for publication as a short communication.